# Application of a 3D-Printed Writing–Typing Assistive Device in Patients with Cervical Spinal Cord Injury

**Ji Hwan Kim [1], Hee Seung Yang [1,\*], Seung Hyun Han [1], Byung Min Lee [1], Youn Kyung Lee [1], Woo Sob Sim [2], Gwan Su Park [2], Seul Bin Na Lee [2] and Min Jo [2]**

[1]   Department of Physical Medicine and Rehabilitation, Veterans Health Service Medical Center, Seoul 05368, Korea

[2]   Center of Prosthetics and Orthotics, Veterans Health Service Medical Center, Seoul 05368, Korea

\*   Correspondence: yang7310@naver.com; Tel.: +82-2-2225-1397; Fax: +82-2-2225-1698

**Abstract:** The current study aimed to assess the effects of a customized three-dimensional (3D) printed writing and typing assistive device in patients with cervical spinal cord injury who presented with severe hand dysfunction. Three patients with cervical spinal cord injury who presented with severe hand dysfunction were included in the analysis. The patients' writing and typing abilities were evaluated after using the silicon assistive device made from a 3D-printed frame for 4 weeks. Patient discomfort and issues were evaluated. Customized 3D-printed writing and typing assistive devices were developed. The Korean Western Aphasia Battery (K-WAB), particularly the writing part, and the word practice program of Hangeul were utilized to assess device effects. All patients with cervical spinal cord injuries (SCIs) performed writing or typing using a customized assistive device. Patients 2 and 3 had better typing and writing accuracies based on the word practice program of Hangeul and the K-WAB, respectively. However, patient 3 had increased time, which was associated with the process of adapting to the use of the customized device. Nevertheless, he was highly satisfied with the device. The patient's typing and writing speed and accuracy improve with the customized 3D-printed device, which can lead to a better performance in the activities of daily living.

**Keywords:** three-dimensional printing; assistive devices; spinal cord injury; hand function; tetraplegia

## 1. Introduction

Spinal cord injury (SCI), particularly cervical injury, is a major disorder that causes sensory, motor, and autonomic dysfunctions in different body parts served by the spinal cord below the level of injury [1]. It often leads to permanent sequelae and eventually results in extensive physical and emotional disabilities among patients, families, and society [1]. However, currently, there is no treatment that can promote the complete recovery of functional status in patients with SCI [2]. In particular, patients with tetraplegia cannot perform all the activities of daily living (ADL) due to arm and hand dysfunction. Hence, they require assistance from others. ADL assistive devices can improve the independence of patients with cervical SCI and reduce caregiver burden. Therefore, it is important to prescribe an assistive device for cervical SCI with consideration of the person's muscle strength, residual function, and individual circumstances [3]. However, an assistive device made from ready-made products is manufactured in a batch shape and size. Hence, it is difficult to match to the individual characteristics of patients. Prefabricated aids and splints can be uncomfortable and unsuitable. Thus, they can cause different conditions such as pain, edema, pressure, and perspiration, and they cannot maximize patient function. Therefore, patients with cervical SCI cannot gain maximum independence.

Wäckerlin et al. investigated the self-reported availability and the unmet needs of 150 patients with tetraplegia who had impaired hand function and who used 18 different assistive devices [4]. In this study, 67.3% of patients with tetraplegia did not use any

assistive device to compensate for impaired hand function. One or two devices were available for 10.0% and 10.7%, respectively [4]. The writing and typing devices were only available in 10.6% and 14.1% of patients, respectively [4]. Therefore, there is a need for a writing and typing devices customized according to SCI severity. In the rehabilitative fields, there is a growing interest in the development of customized assistive devices to accommodate the body's interpersonal anatomical transformation [5]. Three-dimensional (3D) printing offers an innovative method for therapists or physicians to conveniently make customized assistive devices for special use in clinical settings. Three-dimensional printing is a computer-aided manufacturing method that can create 3D objects using different materials such as plastic, metal, liquids, and even living cells [6,7]. Recently, there is a tendency to make the lower limb aesthetic covers, insoles, silicone liners, wheelchair control devices, and spinal orthoses to suit disability characteristics and individual body shape by using 3D design and scanning. This technology is beneficial as it is cost-effective, can be customized, and has enhanced productivity. Hence, it has attracted significant attention in the biomedical field [8,9].

The current study aimed to evaluate the effects of customized 3D-printed writing and typing assistive devices in patients with cervical SCI who presented with severe hand dysfunction. Accordingly, it was applied to patients with chronic SCI, and its effects on hand function were evaluated via clinical tests. The device aims to improve the quality of life and self-esteem of patients with cervical SCI.

## 2. Materials and Methods

The current study included patients admitted to the rehabilitation medicine department due to tetraplegia caused by cervical SCI from June 2019 to November 2019. The inclusion criteria were as follows: patients with normal cognitive functions who had severe hand dysfunction and who cannot perform normal ADL, and those without communication issues. Patients with tetraplegia caused by traumatic brain injury or stroke were excluded.

To develop the special assistive device, the patient's hand was initially positioned for specific functions and was scanned with a 3D scanner. The scanned 3D image was converted into STL files, which were loaded or designed using Autodesk Fusion 360 (San Rafael, CA, USA). Then, 3D printing was performed using RAISE3D Pro2 Plus (Tesla, Irvine, CA, USA) with the fused deposition modeling method (FDM), which involved stacking up melted flexible thermoplastic elastomer filaments layer by layer. FDM is one of the most widely used manufacturing techniques in 3D printing. Moreover, it has several advantages: it can print products rapidly and is cost-effective [8]. Polylactic acid (PLA), which is the same thermoplastic material used in manufacturing conventional orthosis, is the plastic monofilament used in the hand device. We manufactured and applied various types of 3D-printed assistive devices.

We used this 3D printer technology to design a customized orthosis for the size of the patient's hand. At the beginning of the study, we designed a prototype made of PLA material and applied it to the patients. Figure 1 summarizes the design evolution of various device prototypes for the recruited patients. First, is an assistive device made of PLA material wherein the patients place their finger (Figure 1A,B). However, patients found it difficult to maintain pronation and were unable to continuously use the assistive device. As such, a device that can be used in the neutral position and allow writing and typing was needed. The second device designed allowed for typing and writing even in a neutral position but had the disadvantage of having to change the device, which was cumbersome depending on the situation. Another disadvantage was that the pen fell out when writing (Figure 1C). Accordingly, each function was separately produced using a design that allows for simultaneous typing and writing. Moreover, unlike the second design, one fixing screw was added (Figure 1D). However, one disadvantage of this design was problems in both the typing device and the integrated device used by inserting the patients' fingers. In other words, several patients with spinal cord injury, especially those with cervical injuries, exhibited hand muscle atrophy, which made it difficult to utilize the device given the

difference in the individual's hand size or hand function. In summary, plastic-type typing and writing assisted devices could not be applied because the hand grasp power of patients with cervical SCI is extremely weak. Further, plastic-type devices were extremely hard and difficult to carry.

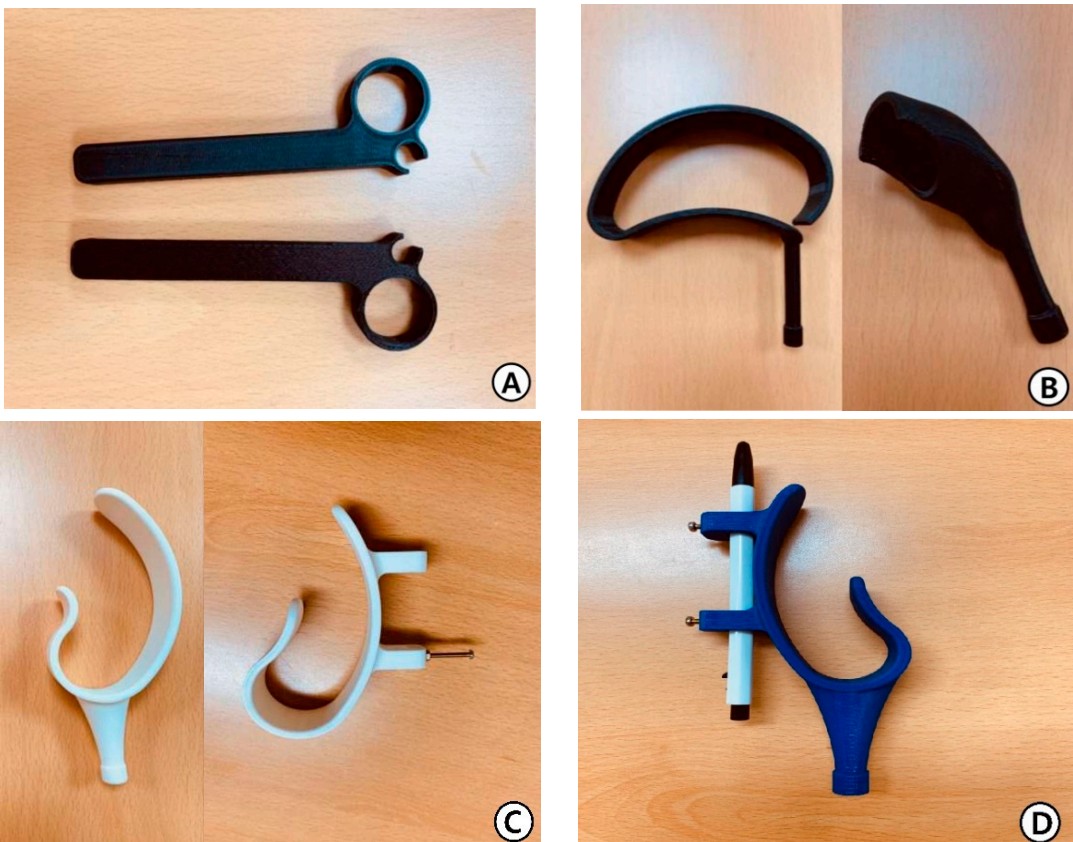

**Figure 1.** (**A**) First 3D-printed writing device. (**B**) First 3D-printed typing device. (**C**) Second 3D-printed writing (**left**) and typing (**right**) device. (**D**) Third 3D-printed typing and writing integration device.

Thus, we switched to making and applying band-type assistive devices using silicon. A frame was designed for band-shaped devices (Figure 2B). Then, it was printed using RAISE3D Pro2 Plus (Tesla, Irvine, CA, USA) with FDM (Figure 2C). Liquid silicone of different hardness was poured into the 3D-printed frame (Figure 2E,F). The liquid silicon comprised materials A and B. The materials were mixed at room temperature in a 1:1 ratio (A, which is the main material, and B, which is the hardener material and is cured for 16 h). The sizing scheme was based on the hand measurements of each patient (Figure 2D). Thus, we created a device that was pulled out with a 3D-printed frame of different sizes that was suitable for each patient and that can improve function (Figure 2E).

The time required for typing and writing before and after wearing the device was evaluated by an occupational therapist. Typing evaluation was conducted using Hangeul and other computerized word practice programs of Hangeul and Computer Corporation (HANCOM Inc., Seoul, Korea) that are used to measure typos, accuracy, and time requirement. Hangeul, a Korean typing evaluation program, checks the number of errors in each syllable when typing a given sentence, displays the accuracy as a percentage, and converts the time taken while typing into typing speed. In this program, all patients were given the same sentence consisting of 6 Korean words with 14 syllables. Moreover, typing and writing were evaluated by selectively using the writing part of the Korean Western Aphasia Battery (K-WAB), consisting of naming, number writing, dictation, and copying [10]. The

naming part asked the patient to write his or her own name. The number writing part required patients to write the numbers in two ways, once using Arabic numbers from 1 to 10 and then once in Korean (e.g., one, two . . . nine, and ten). Thereafter, they were asked to list the days of the week from Monday to Sunday. The dictation part required patients to dictate a word composed of one to four syllables, making sure to emphasize each syllable. In the copying and writing part, patients were requested to exactly copy a presented sentence consisting of 10 words and 17 syllables. Each time the patient performed an item, the duration needed to complete each item was measured in seconds. All study-related procedures were performed in accordance with the ethical standards of the institutional and/or national research committee and the 1964 Declaration of Helsinki. The current study was approved by the institutional review board of Veterans Health Service Medical Center (no. 2019-03-025).

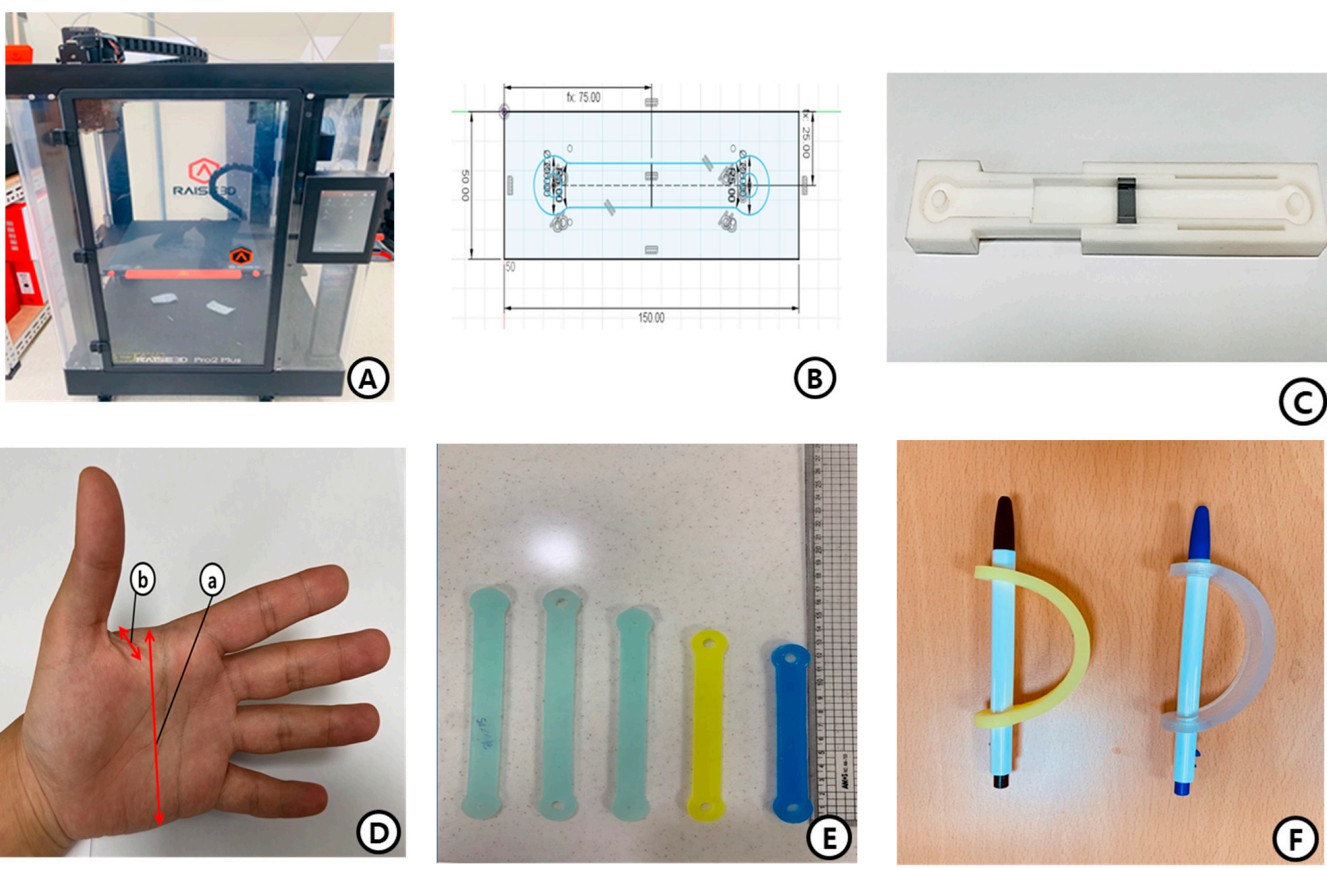

**Figure 2.** (**A**) Three-dimensional printer (RAISE3D Pro2 Plus; Tesla, Irvine, CA, USA). (**B**) Frame design on the STL file. (**C**) Three-dimensional-printed polylactic acid frame using the fused deposition modeling method. (**D**) a. Handbreadth from the MCP joint of the thumb to the MCP joint of the 2nd finger along the surface of the hand. b. Width between the index and little finger. (**E**) Five silicon assistive devices according to length. (**F**) Printing typing and writing integration device.

Typing and writing accuracy and speed were evaluated before and after using the devices made from a 3D-printed mold for 4 weeks. The patients were taught how to use the assistive devices only once by the occupational therapist. If the patient wanted the device, it was used during treatment and in ADL.

## 3. Results

All patients completed the study, and the results are summarized in Tables 1–3.

### 3.1. Patient 1

A 73-year-old male patient presented with tetraplegia (Asia Impairment Scale (AIS) A) due to SCI. Muscle contraction below the C6 level was impaired. The patient had 0 grade on both hands. Hence, he required assistance from a caregiver in writing his name. Moreover, the patient could not write normally. Thus, evaluation could not be performed. However, after the device was worn (Figure 3A), all evaluation items could be performed (Table 3). The patient did not want to be evaluated because he did not use a computer and did not require a typing aid (Tables 1 and 2).

### 3.2. Patient 2

A 46-year-old male patient presented with tetraplegia (AIS A) due to SCI. Muscle contraction was impaired below the C7 level. The patient had a poor grade on both hands. He could catch objects at an extremely slow speed. At first, the assistive device of PLA formulations was used. However, the device could not be worn due to atrophy. Therefore, a silicone device that was manufactured based on the size of each hand was developed and used (Figure 3B). The keyboard typing accuracy and speed increased, and both writing and typing improved after wearing the orthosis (Tables 1–3).

**Table 1.** Results based on the word practice program of Hangeul.

| Program | | Pre-Test | Post-Test |
|---|---|---|---|
| Patient 2 | Error [1] (n) | 3 | 0 |
| | Accuracy [2] (%) | 78 | 100 |
| | Time (s) | 58.38 | 51.97 |
| Patient 3 | Error (n) | 0 | 0 |
| | Accuracy (%) | 100 | 100 |
| | Time (s) | 42.53 | 43.43 |

[1] The number of syllables with typos; [2] The percentage of correctly typed syllables among the total.

**Table 2.** Results based on the typing K-WAB score.

| Typing K-WAB | | Pre-Test | | Post-Test | |
|---|---|---|---|---|---|
| | | Score | Time (s) | Score | Time (s) |
| Patient 2 | Name | 2 | 4.93 | 2 | 4.81 |
| | One–Ten | 10 | 44.40 | 10 | 38.25 |
| | 1–10 | 5 | 10.50 | 5 | 5.69 |
| | Monday to Sunday | 7 | 31.75 | 7 | 19.9 |
| | Dictation (one letter) | 1 | 2.75 | 1 | 3.09 |
| | Dictation (two letters) | 2 | 4.60 | 2 | 4.06 |
| | Dictation (three letters) | 3 | 8.57 | 3 | 5.34 |
| | Dictation (four letters) | 4 | 5.56 | 4 | 5.85 |
| | Copy and write | 10 | 59.66 | 10 | 4.81 |
| Patient 3 | Name | 2 | 5.50 | 2 | 7.50 |
| | One–Ten | 10 | 39.78 | 10 | 34.32 |
| | 1–10 | 5 | 5.18 | 5 | 5.53 |
| | Monday to Sunday | 7 | 18.31 | 7 | 14.22 |
| | Dictation (one letter) | 1 | 1.75 | 1 | 1.69 |
| | Dictation (two letters) | 2 | 3.44 | 2 | 3.34 |
| | Dictation (three letters) | 3 | 4.75 | 3 | 4.06 |
| | Dictation (four letters) | 4 | 3.87 | 4 | 3.66 |
| | Copy and write | 10 | 37.28 | 10 | 43.43 |

**Table 3.** Results based on the writing K-WAB score.

| Writing K-WAB | | Pre-Test | | Post-Test | |
|---|---|---|---|---|---|
| | | Score | Time (s) | Score | Time (s) |
| Patient 1 | Name | 0 | UC | 2 | 53.09 |
| | 1–10 | 0 | UC | 10 | 238.50 |
| | 1–10 | 0 | UC | 5 | 55.22 |
| | Months to days | 0 | UC | 7 | 86.78 |
| | Dictation (one letter) | 0 | UC | 1 | 11.63 |
| | Dictation (two letters) | 0 | UC | 2 | 20.78 |
| | Dictation (three letters) | 0 | UC | 3 | 20.97 |
| | Dictation (four letters) | 0 | UC | 4 | 30.78 |
| | Copy and write | 0 | UC | 10 | 210.37 |
| Patient 2 | Name | 2 | 12.59 | 2 | 12.59 |
| | One-Ten | 10 | 68.41 | 10 | 52.31 |
| | 1–10 | 5 | 13.16 | 5 | 11.63 |
| | Monday to Sunday | 7 | 21.90 | 7 | 16.63 |
| | Dictation (one letter) | 1 | 3.82 | 1 | 3.50 |
| | Dictation (two letters) | 2 | 5.78 | 2 | 5.69 |
| | Dictation (three letters) | 3 | 7.79 | 3 | 8.22 |
| | Dictation (four letters) | 4 | 6.97 | 4 | 7.81 |
| | Copy and write | 10 | 47.75 | 10 | 55.19 |
| Patient 3 | Name | 2 | 9.47 | 2 | 9.97 |
| | One-Ten | 10 | 52.78 | 10 | 59.25 |
| | 1–10 | 5 | 14.91 | 5 | 15.50 |
| | Monday to Sunday | 7 | 21.49 | 7 | 23.97 |
| | Dictation (one letter) | 1 | 3.91 | 1 | 4.12 |
| | Dictation (two letters) | 2 | 5.03 | 2 | 5.56 |
| | Dictation (three letters) | 3 | 8.50 | 3 | 9.53 |
| | Dictation (four letters) | 4 | 8.06 | 4 | 9.19 |
| | Copy and write | 10 | 61.60 | 10 | 60.87 |

*3.3. Patient 3*

A 34-year-old male patient presented with tetraplegia (AIS A) due to SCI. Muscle contraction was impaired below the C5 level. The patient had poor grade on both hands, and he could catch objects at an extremely slow speed. First, the patient used an assistive device made of PLA material, and it could be used by placing a finger. However, the patient had difficulties in maintaining pronation. Thus, the device's design was modified to help the patient assume the neutral position and perform typing (Figure 1C,D). Only patient 3 was evaluated by applying and typing as a PLA formulations device. The patient's typing speed was slightly slower with the device (Tables 1–3). However, his typing accuracy improved (Figure 3D).

Patient 3 claimed that the PLA formulation device was inconvenient because he mainly used his smartphone and rarely a computer. When using a smartphone, there was no problem using it without device. Moreover, he stated that "silicon devices are extremely convenient when used in writing" and "it is convenient to use not only pens but also other objects (e.g., toothbrush) in between them" (Figure 2F). The keyboard typing and writing accuracies were almost similar before and after wearing the device, and the overall speed was slightly slower. The patient could type and write to some extent even without the assistive device, and the process of adapting to the use of the new devices was slow.

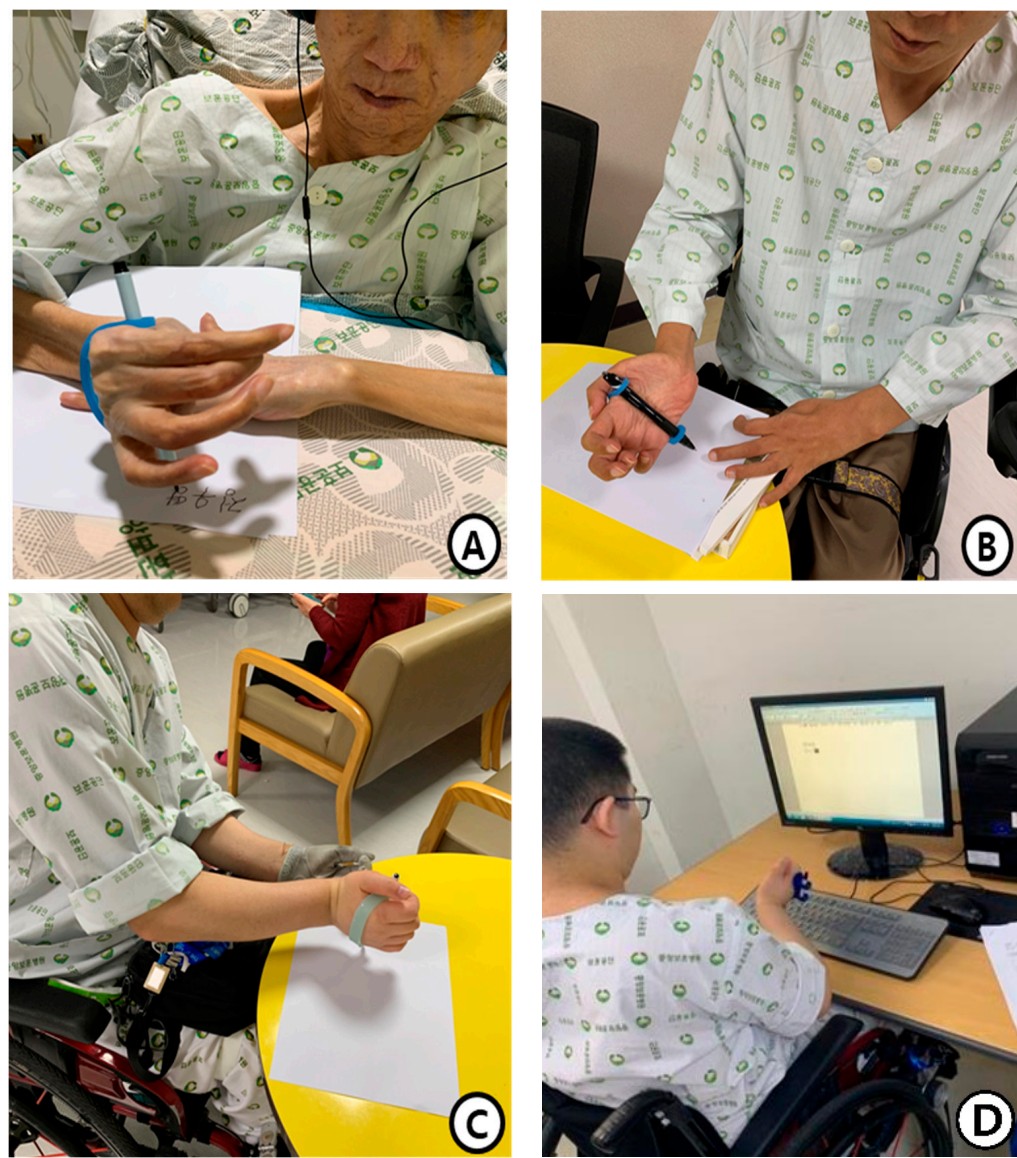

**Figure 3.** (**A**) Patient 1 wearing the fourth three-dimensional (3D)-printed writing integration device made of silicone material. (**B**) Patient 2 wearing the 3D-printed typing and writing integration device made of silicone material. (**C**) Patient 3 wearing the fourth 3D-printed typing and writing integration device made of silicone material. (**D**) Patient 3 wearing the 3D-printed typing integration device with the polylactic acid material.

## 4. Discussion

With the rapid development of CAD, 3D scanning, and 3D printer capabilities, applications in the field of 3D-printed rehabilitation devices have a tremendous potential [11]. Moreover, 3D printing allows you to establish assistive orthopedic devices that are more cost-effective and durable, lighter, and easier to replace than conventional braces [12]. In this study, customized 3D-printed writing and typing assistive devices were shown to improve the quality of life of patients during writing and typing tasks. The current study showed that patients who could not write were able to do so, and their accuracy and speed improved. Our results also showed time delays in the examination after the use of assistive device, although we considered this to have resulted from low proficiency. Regardless of the outcomes, all patients had a high subjective satisfaction, which led to psychiatric stress relief and ADL improvement. Thus, the device was effective in improving the quality of life. Assistive devices created using 3D printing technology has several advantages.

Among them, the most significant is that these devices can be easily transformed for specific purposes based on basic hand shapes using stored 3D image files. Although it was not explained in detail, different plastic hand aids were developed according to the basic hand forms. Then, they were applied to the three patients. However, the use of these devices was not successful due to poor utilization and patient satisfaction. Hence, the silicon writing and typing device made from a 3D-printed frame was established, and the thickness and length of each patient's hand were measured to make the device more suitable for each patient.

Device cost is a hindrance for the application of devices in patients with severe disability who require orthosis. Nonetheless, Alireza Mohammadi et al. developed soft robotics prosthetic hands with a relatively low price at 200 USD. However, this does not include the quick disconnect wrist [13]. Therefore, we developed a low-cost and lightweight 3D-printed hand assistive device using a simple design. Therefore, the patient's compliance would be higher, which might play an important role in improving function. To help more patients with physical disabilities, low-cost and simple assistive devices are essential in the rehabilitation field. Hence, these 3D-printed products may be cost-effective, and their use as an alternative to conventional devices is promising.

The current study had several limitations. Patients were taught how to use the writing and typing device. However, they were not trained. Then, they were evaluated after self-use. Moreover, only three patients were included in the study, and long-term follow-up was not performed. Although the long-term usability of devices was not assessed, users could benefit more from orthosis if they had a sufficient training time to adapt to it. Nevertheless, further research should be performed to investigate the feasibility of this technology as it might be an alternative to the conventional orthosis in patients with cervical SCI. Finally, this study did not objectively evaluate patient satisfaction. Hence, future research should include an objective evaluation of satisfaction in addition to improved performance.

## 5. Conclusions

Customized 3D-printed writing and typing devices may improve the writing and typing abilities of patients with severe disabilities, thereby promoting greater satisfaction. We believe that this study will be the cornerstone of research on assistive devices developed using the alternative 3D printing technology of conventional orthosis in patients with cervical SCI who have tetraplegia.

**Author Contributions:** Conceptualization, J.H.K. and H.S.Y.; Data curation, J.H.K. and Y.K.L.; Formal analysis, J.H.K.; Investigation, S.H.H. and B.M.L.; Methodology: H.S.Y.; Project administration: H.S.Y.; Resources: H.S.Y.; Software: W.S.S., G.S.P., S.B.N.L. and M.J.; Writing—original draft preparation, J.H.K.; Writing—review and editing, H.S.Y.; Visualization, J.H.K.; Supervision, H.S.Y. All authors have read and agreed to the published version of the manuscript.

**Funding:** This research received no external funding.

**Institutional Review Board Statement:** The study was conducted in accordance with the Declaration of Helsinki, and approved by the Institutional Review Board of the Veterans Health Service Medical Center (2000-00-000-000).

**Informed Consent Statement:** Informed consent was obtained from all subjects involved in the study. Written informed consent has been obtained from the patients to publish this paper.

**Data Availability Statement:** Data will be made available on reasonable request from the corresponding author.

**Conflicts of Interest:** The authors declare no conflict of interest.

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
