# Peer review of "Application of a 3D-Printed Writing–Typing Assistive Device in Patients with Cervical Spinal Cord Injury"

_applsci, doi:10.3390/app12189037_

Round 1

Reviewer 1 Report

1. Present title
"Application of a Three-Dimensional-Printed Writing and Typing Assistive Device in Patients with Cervical Spinal Cord Injury"

Will the following change be better?

"Application of a 3D-printed Writing-Typing Assistive Device in the Patients having Cervical Spinal Cord Injury"

2. Although the structural design of the prototype (Figure 1) is quite intuitive,
still if you can discuss a design methodology to finalize the structural shape/geometry of the assistive device, that would have been helpful.

3. The Results section starts with the following statement
" All patients completed the study, and the results are summarized in Tables 1, 2, and 3."

  A detailed discussion on the several parameters mentioned in Table 1,2 will be useful before mentioning the tables.

  4.  What do you mean by accuracy (%)/error (%) here? Explain. This is normally better expressed by confusion matrix.

 5.  What do you mean by "time" in Table 1 , 2? because the time can vary with words, sentences and the time to perceive them.

6. In page 4, line 118, "Typing and writing accuracy and speed and patient satisfaction were evaluated be-"
How have you evaluated/measured patient satisfaction ? Normally "patient satisfaction" measurement comes under cognitive load analysis, which often takes the help of EEG signals. How have you measured the satisfaction parameter? 

Reviewer 2 Report

This article aims at the cervical spinal cord injury patients with severe hand dysfunction and uses the 3D printing writing and typing assistant device. This research has certain practical value.

A list of comments and advices can be found as follows:

1. The serial number of the title is incorrectly labeled, Patient 3 should be 3.3.

2. It is recommended that more patients be included in the study.

3. Some necessary theoretical analyses are suggested to be supplemented in methods.

Round 2

Reviewer 1 Report

The authors have addressed the comments. The article may be accepted.